# A Zika Vaccine Generated Using the Chimeric Insect-Specific Binjari Virus Platform Protects against Fetal Brain Infection in Pregnant Mice

**DOI:** 10.3390/vaccines8030496

**Published:** 2020-09-02

**Authors:** Jessamine E. Hazlewood, Daniel J. Rawle, Bing Tang, Kexin Yan, Laura J. Vet, Eri Nakayama, Jody Hobson-Peters, Roy A. Hall, Andreas Suhrbier

**Affiliations:** 1Inflammation Biology Group, QIMR Berghofer Medical Research Institute, Brisbane, QLD 4006, Australia; Jessamine.Hazlewood@qimrberghofer.edu.au (J.E.H.); Daniel.Rawle@qimrberghofer.edu.au (D.J.R.); Bing.Tang@qimrberghofer.edu.au (B.T.); Kexin.Yan@qimrberghofer.edu.au (K.Y.); 2School of Chemistry and Molecular Biosciences, University of Queensland, St. Lucia, QLD 4072, Australia; l.vet@uq.edu.au (L.J.V.); j.peters2@uq.edu.au (J.H.-P.); 3Department of Virology I, National Institute of Infectious Diseases, Tokyo 162-0052, Japan; nakayama@nih.go.jp; 4Australian Infectious Diseases Research Centre, Brisbane, QLD 4006, Australia

**Keywords:** vaccine, Binjari, Zika, dengue, mouse model

## Abstract

Zika virus (ZIKV) is the etiological agent of congenital Zika syndrome (CZS), a spectrum of birth defects that can lead to life-long disabilities. A range of vaccines are in development with the target population including pregnant women and women of child-bearing age. Using a recently described chimeric flavivirus vaccine technology based on the novel insect-specific Binjari virus (BinJV), we generated a ZIKV vaccine (BinJ/ZIKA-prME) and illustrate herein its ability to protect against fetal brain infection. Female IFNAR^−/−^ mice were vaccinated once with unadjuvanted BinJ/ZIKA-prME, were mated, and at embryonic day 12.5 were challenged with ZIKV_PRVABC59_. No infectious ZIKV was detected in maternal blood, placenta, or fetal heads in BinJ/ZIKA-prME-vaccinated mice. A similar result was obtained when the more sensitive qRT PCR methodology was used to measure the viral RNA. BinJ/ZIKA-prME vaccination also did not result in antibody-dependent enhancement of dengue virus infection or disease. BinJ/ZIKA-prME thus emerges as a potential vaccine candidate for the prevention of CSZ.

## 1. Introduction

Zika virus (ZIKV) is the etiological agent of congenital Zika syndrome (CZS), which encompasses a range of birth defects associated with ZIKV infection of the mother during pregnancy and transplacental infection of the fetus. CZS is primarily (but not exclusively) associated with neurological abnormalities such as microcephaly that can lead to complex and life-long disabilities [1,2]. The WHO declared ZIKV a Public Health Emergency of International Concern from February to November 2016, with a range of vaccines against ZIKV currently in development [3]. As pregnant women and/or women of child bearing age are the key targets for vaccination, heightened safety concerns might lead to a preference for inactivated, subunit, or modified RNA vaccines rather than live, live-attenuated, or live virus-vectored vaccines [4,5,6,7,8]. Use of vaccine adjuvants during pregnancy may also raise safety concerns [5,6].

Live-attenuated chimeric flavivirus vaccines are well established and generally use an attenuated flavivirus vaccine as a backbone, with the latter’s pre-membrane and envelope coding sequence (*prME*) replaced with the *prME* coding sequence of the target flavivirus, for instance, ZIKV *prME* with a licensed Japanese encephalitis live-attenuated vaccine as the backbone [8], or *prME* from dengue viruses with the licensed Yellow fever 17D live-attenuated vaccine as the backbone [9]. We recently applied this concept to the generation of chimeric flavivirus vaccines using Binjari virus (BinJV) [10]. BinJV is a lineage two, insect-specific flavivirus isolated from *Aedes normanensis* mosquitoes captured in the Northern Territory, Australia [11]. BinJV emerged to be remarkably tolerant of substitution of its *prME* genes with the *prME* genes of flaviviruses that cause disease in humans. The resulting chimeras have potential utility as vaccines as they retained their ability to replicate in insect cells (allowing vaccine manufacture), but are unable to replicate in vertebrate cells (providing complete attenuation in the vaccine recipient) [10]. The BinJV platform has now been used to generate a number of flavivirus vaccines that have induced protective immunity in mouse models; these include a chimeric Yellow fever (YF) virus vaccine, BinJ/YF-prME [12]; a chimeric West Nile (WN) virus vaccine, Binj/WN-prME [13]; and a chimeric ZIKV vaccine, BinJ/ZIKA-prME [10].

BinJ/ZIKA-prME vaccination, in the absence of adjuvant, was previously shown to protect against viremia in male and female IFNAR^−/−^ mice and to protect against testicular damage in male IFNAR^−/−^ mice [10]. However, BinJ/ZIKA-prME has not been evaluated for its ability to protect against fetal brain infection, clearly a key objective of a ZIKV vaccine [3,14]. Protection against fetal brain infection may represent a more onerous requirement for a ZIKV vaccine, with a contention arguing that induction of high levels of neutralizing antibodies may be needed to provide sterilizing immunity [15,16,17,18,19]. The brain is often considered to be an immune privileged site [20], so any transplacental infection reaching the fetal brain may therefore arguably replicate and cause brain damage, largely unimpeded by maternal immunity [16,21].

Because of serological cross-reactivity between ZIKV and dengue virus (DENV), an additional requirement for ZIKV vaccines is that vaccine-induced responses should not induce antibody-dependent enhancement (ADE) of DENV infections [22,23]. Whether naturally acquired anti-ZIKV antibodies could cause ADE of DENV infection with clinically important consequences in humans is currently unclear [24], although a recent study of pediatric patients in Nicaragua supports the contention [25]. ADE of DENV infection by ZIKV antibodies has also been shown in mice and non-human primates [26,27]. The reverse, anti-DENV antibodies causing ADE of ZIKV, would appear not to be associated with exacerbated ZIKV infections in humans [28]. Either way, regulators would likely seek assurances that a new ZIKV vaccine does not cause ADE of DENV because of some unforeseen suboptimal feature of vaccine design or production. DENV infection during pregnancy can cause severe maternal complications, and severe maternal infections may also be associated with adverse fetal outcomes [29]. Herein, we show that BinJ/ZIKA-prME vaccination protects against viremia and fetal brain infection in IFNAR^−/−^ dams, and that such vaccination does not cause DENV ADE in IFNAR^−/−^ mice.

## 2. Materials and Methods

### 2.1. Ethics Statement

Mouse work was undertaken in accordance with the Australian Code for Care and Use of Animals for Scientific Purposes, as outlined by the National Health and Medical Research Council of Australia. Animal work was approved by the QIMR Berghofer Medical Research Institute Animal Ethics Committee (Approval: A1604-611M).

### 2.2. ZIKV_PRVABC59_

An infectious clone of ZIKV_PRVABC59_ (GenBank ID: MH158237.1) [30] was kindly provided by Dr S. Tajima (Department of Virology I, National Institute of Infectious Diseases, Tokyo, Japan). ZIKV_PRVABC59_ was recovered by transfection of Vero E6 cells and virus stocks were generated in C6/36 cells. Viral stocks were checked for mycoplasma and endotoxin [31,32].

### 2.3. Vaccination and Antibody Responses

The chimeric BinJ/ZIKA-prME vaccine was generated using a modified circular polymerase extension reaction methodology and used the ZIKV_Natal_
*prME* genes [10]. BinJ/ZIKA-prME and BinJV (Genbank ID: MG5870308) were grown in C6/36 cells (ATCC CRL-1660) and were purified using polyethylene glycol precipitation, a sucrose cushion and a potassium tartrate gradient as described previously [10]. Protein concentrations were determined using Bradford assay (Bio-Rad Laboratories, Hercules, CA, USA). IFNAR^−/−^ mice on a C57BL/6J background were provided by Prof P. Hertzog (Monash University, Melbourne, VIC, Australia) [33] and were bred in-house at QIMR Berghofer. Female IFNAR^−/−^ mice (2–4 months of age) were anaesthetized and vaccinated i.m. into both quadriceps muscles (50 µL per muscle), with 10 µg of BinJ/ZIKA-prME or BinJV delivered per mouse. Control mice received 50 µL of PBS. Serum IgG ELISA titers (using purified ZIKV_MR766_ antigen) and neutralization assays (using ZIKV_PRVABC59_) were undertaken as described previously [10].

### 2.4. ZIKV Challenge of Pregnant IFNAR^−/−^ Mice

ZIKV_PRVABC59_ challenge of pregnant dams (*n* = 20) was undertaken as described [14] with some modifications. Briefly, vaccinated (*n* = 6) and control (*n* = 4/5 per group) female IFNAR^−/−^ mice were paired with IFNAR^−/−^ males and when a plug was detected, this was deemed embryonic day 0.5 (E0.5). Weight gain was used to confirm pregnancy; in the absence of weight gain, mice were repaired with males. At E12.5, dams were challenged subcutaneously with 10^4^ CCID_50_ of ZIKV_PRVABC59_ (GenBank ID: KU501215). Mice were bled daily for 5 days, with day 5 post challenge equivalent to E17.5. The mice were then euthanized using CO_2_, and placenta and fetal heads were harvested and stored at −80 °C (for viral titrations) or RNAlater (Qiagen, Hilden, Germany) for qRT-PCR. ZIKV titers in sera and tissues were determined by CCID_50_ assay as described previously using titration on C6/36 cells and virus detection by cytopathic effects in Vero E6 cells [12,14,34], with one modification; supernatants from homogenized tissues were titered using five-fold serial dilutions.

### 2.5. Real Time Quantitative RT-PCR (qRT PCR)

Tissues were transferred from RNAlater to TRIzol (Life Technologies, Carlsbad, CA, USA) with subsequent homogenization of tissue, RNA extractions, and cDNA synthesis undertaken as described previously [12]. qRT PCR was performed on cDNA from maternal placenta, fetal brain, and deformed fetal tissue as described [14] with minor modifications. ZIKV E primers were ZIKV F: 5’-CCG CTG CCC AAC ACA AG-3’; ZIKV R: 5’-CCA CTA ACG TTC TTT TGC AGA CAT-3’ [35], with normalization to murine RPL13A [36]. Cycling conditions were 95 °C for 2 min, then 95 °C for 5 min, and 60 °C for 30 s for 40 cycles. Melt curves were determined over 65 °C to 95 °C with increments of 0.5 °C for 5 s.

### 2.6. Model of DENV ADE

A DENV-D220 female IFNAR^−/−^ ADE mouse model has been described previously, with ADE (mediated by prior injection of an anti-flavivirus E specific antibody) manifesting as significantly increased weight loss and mortality post DENV infection [19]. Female IFNAR^−/−^ mice (4 to 24 weeks old, *n* = 6 per group with a similar age distribution in each group) were anaesthetized and vaccinated once with either 2 µg BinJ/ZIKA-prME, 20 µg BinJ/ZIKA-prME, or PBS (in 50 µL) injected into each *quadriceps femoris* muscle. Mice were infected s.c. with 10^6^ CCID_50_ DENV-220 (Genbank ID: HQ541799) 7 weeks post vaccination. Sera was collected daily for 7 days for viremia analysis and mice were weighed at the indicated times.

### 2.7. Statistical Analysis

IBM SPSS Statistics for Windows, Version 22.0 was used for statistical analyses. The *t*-test was used where the difference in variance was <4, skewness was >2, and kurtosis was <2. Otherwise the Mann Whitney U test was used if the difference in variance <4, and the Kolmogorov-Smirnov test or the Kruskal-Wallis tests were used if the difference in variance >4. For weight change over time, the repeat measures ANOVA was used.

## 3. Results

### 3.1. BinJ/ZIKA-prME Vaccination and Protection against ZIKV Challenge

The chimeric BinJ/ZIKA-prME vaccine (Figure 1a) was generated using a modified circular polymerase extension reaction, and both BinJ/ZIKA-prME and BinJV were propagated in C6/36 cells and purified using a sucrose cushion and a tartrate gradient [10]. Purified BinJ/ZIKA-prME or BinJV were used to vaccinate female IFNAR^−/−^ mice i.m., with control mice mock vaccinated with PBS. Antibody responses were determined 5 weeks post vaccination (Figure 1b). BinJ/ZIKA-prME vaccination induced significant ZIKV-specific IgG ELISA and ZIKV neutralizing antibody responses, with no ZIKV-specific antibodies detected after BinJV or PBS vaccination (Figure 1c,d). Timed matings were initiated at 1 month post vaccination, and after pregnancy was confirmed by weight gain, dams were challenged with ZIKV_PRVABC59_ at E12.5 (Figure 1b). BinJ/ZIKA-prME vaccinated dams showed no detectable viremia, with BinJV vaccinated dams showing a slightly, but not significantly, reduced viremia when compared with the PBS control group (Figure 1e). A single vaccination with BinJ/ZIKA-prME was thus able to provide complete protection against detectable viremia in IFNAR^−/−^ mice, consistent with previous studies [10].

### 3.2. Overt Pregnancy Outcomes after ZIKV Challenge

After ZIKV challenge, dams were euthanized at E17.5 and placental and fetal tissue harvested (Figure 1b). Numbers and visual examination of placenta and fetuses are described in Table 1, with no significant differences in fetal abnormalities emerging between the groups. Deformed fetal-placental masses and intra-uterine growth restriction (IUGR) induced by ZIKV infection have been described previously [14,37]; however, such phenomena can clearly also occur in the absence of ZIKV infection (Table 1) [38,39]. Placentas with no discernable fetus (presumably reabsorbed) were also occasionally observed (Table 1).

### 3.3. BinJ/ZIKA-prME Protected against Fetal Weight Loss and Infection of Placenta and Fetuses

Placenta and fetuses were harvested at E17.5 (5 days post challenge). Outwardly normal fetuses from BinJ/ZIKA-prME vaccinated dams were slightly but significantly heavier (a mean ≈1.1 fold; *p* = 0.008, *t* test) than such fetuses from the PBS vaccinated group (Figure 2a). The same comparison for BinJ/ZIKA-prME and BinJV vaccinated dams was also significant (≈1.09 fold, *p* = 0.02, *t* test) (Figure 2a). BinJ/ZIKA-prME vaccination thus prevented reduction in fetal weights induced by ZIKV infection. In humans, birth weights are also often lower for neonates from ZIKV infected mothers [40].

Placental and fetal samples were analyzed for the presence of infectious ZIKV. No infectious virus was detected in any placental or fetal tissues from dams previously vaccinated with BinJ/ZIKA-prME. In contrast, most of the placental and several fetal tissues contained high titers of virus in the BinJV- and PBS-vaccinated groups (Figure 2b). The deformed fetal/placental masses in the BinJ/ZIKA-prME vaccine group were also negative for virus (Figure 2b, green squares). No discernible differences between the BinJV and PBS vaccinated groups was apparent (Figure 2b). BinJ/ZIKA-prME vaccination was thus able to prevent detectable infection of both placenta and fetuses.

### 3.4. Post-Challenge Viral RNA in Placenta and Fetal Heads

The tissue titration had a detection limit of 1.69 log_10_CCID_50_/g and would thus be unable to detect low levels of virus infection. Placentas and fetal tissues taken at E17.5 were therefore also analyzed by qRT-PCR. No ZIKV RNA was detected in the placenta of BinJ/ZIKA-prME-vaccinated mice, except for one equivocal positive that displayed a specific PCR product in 5 out of 11 PCR runs (Figure 2c). All placenta from BinJV and PBS vaccinated groups showed the presence of ZIKV RNA (i.e., Cq value of <40 and a ZIKV-specific melt-curve). A similar picture emerged for fetal heads, although in the BinJV- and PBS-vaccinated groups a small number of fetal heads showed no detectable ZIKV RNA (Figure 2d); qRT PCR and viral titrations used different fetal heads. Deformed fetal/placental masses in the BinJV and PBS vaccinated groups had relatively high levels of viral RNA, but no viral RNA was detected in these tissues from BinJ/ZIKA-prME-vaccinated mice (Figure 2e). Such deformed fetal tissues can arise in the absence of ZIKV infection (Table 1), with such tissues (like fetal heads) clearly also becoming readily infected in dams that have no protective immunity. In summary, except for one equivocal sample, after challenge with ZIKV_PRVABC59_ no viral RNA was detected in placental or fetal tissues in dams vaccinated with BinJ/ZIKA-prME. In contrast, in control vaccinated dams, nearly all placental and fetal tissues contained detectable levels of ZIKV RNA at E17.5.

Statistical comparisons of placenta and fetal head viral RNA levels showed slightly lower (Figure 2c,d), but statistically significant, levels of viral RNA in the BinJV group when compared with the PBS group. For placenta the mean level of viral RNA in the BinJV group was 0.23 ± SD 0.54, compared with 0.65 ± SD 1.0 for the PBS group; *p* = 0.033, Mann-Whitney U test. For fetal heads these values were 0.0001 ± SD 0.003, compared with 0.000127 ± SD 0.00021; *p* = 0.008, Mann-Whitney U test. Although BinJV and ZIKV are very distantly related, a low level of antigenic overlap may explain these findings, with a conserved epitope in the E-protein fusion peptide domain recently identified [11].

### 3.5. BinJ/ZIKA-prME Vaccination Did Not Induce ADE of DENV-2 Infection

A potential issue for ZIKV vaccines is that they (if perhaps poorly designed) might induce ADE of DENV infection. Using a previously established female IFNAR^−/−^ mouse model of DENV-D220 ADE [19], the ability of BinJ/ZIKA-prME vaccination to induce DENV ADE was assessed. IFNAR^−/−^ mice were vaccinated with 2 µg or 20 µg of BinJ/ZIKA-prME, or were mock vaccinated with PBS; mice were then infected with DENV-D220 (Figure 3a). DENV viremia in the 20 µg BinJ/ZIKA-prME was slightly lower than the PBS controls, although this reached significance only on day 3 (Figure 3b). In this model ADE manifests as increased weight loss [19]; however, weight loss in BinJ/ZIKA-prME vaccinated mice was actually significantly reduced when compared with PBS controls (Figure 3c). These results are consistent with the serological cross-reactivity between ZIKV and DENV antibodies [41], but suggests that for BinJ/ZIKA-prME vaccination, this manifests as mild cross-protection against viremia rather than ADE.

## 4. Discussion

We illustrate herein the potential utility of the BinJV technology for the development of a ZIKV vaccine to protect against CZS. A single vaccination with unadjuvanted BinJ/ZIKA-prME was able to prevent fetal brain infection in IFNAR^−/−^ dams challenged with ZIKV. 

Although neutralizing titers upwards of 1/10,000 have been proposed to be needed to prevent viremia and tissue infection [16], herein 50% end point neutralization titers as low as 1/220 titer were associated with prevention of infection of fetal brain tissue, and (with a single possible exception) infection of placenta. Our data are perhaps more in line with a study suggesting 50% end point neutralizing titers of ≥1/460 are required to prevent transplacental transmission and titers of ≥1/430 are required to guarantee that placentas are not infected [42]. However, as neutralization assays are difficult to standardize and behave slightly differently depending on the specific techniques used, only broad comparisons can be made. Exceptionally high vaccine-induced neutralizing titers are not required to protect against congenital brain damage after infection of pregnant cows with bovine viral diarrhea virus (family *Flaviviridae*, genus *Pestivirus*) [43,44,45] or infection of pregnant mothers with rubella virus [46,47]. Our observations thus support the contention that a ZIKV vaccine may protect against fetal brain infection, without the need to induce and maintain exceptionally high levels of neutralizing antibody titers.

Vaccine-induced T cell responses likely also contribute to protection against transplacental infection, although the required phenotypes and their contributions in different model systems remain to be fully elucidated [48]. Induction of T cell responses by the chimeric BinJV vaccines also remains to be characterized, although wild-type mice rather than IFNAR^−/−^ mice are being used for these studies, given the roles of type I IFNs in T cell biology [12,49,50].

The BinJ/ZIKA-prME vaccine encoded the *prME* sequence of ZIKV_Natal_ [10] and protected dams against challenge with another American isolate, ZIKV_PRVABC59_, with ZIKV believed to exist as a single serotype [51]. ZIKV_PRVABC59_ is somewhat more virulent than ZIKV_Natal_ [34], with the mean peak viremia in IFNAR^−/−^ dams for the later ≈1 log lower [14]. In addition, although infection of IFNAR^−/−^ dams at E6.5 with ZIKV_Natal_ allows for normal development of some fetuses [34], such infections with ZIKV_PRVABC59_ resulted in most fetuses being highly deformed or reabsorbed (data not shown). Hence the ZIKV_PRVABC59_ challenge was delayed till E12.5, allowing assessment of fetal brain titers and potentially providing a more robust challenge model for assessing placental titers, given that the maternal viremia had not resolved at this time in control mice. 

The demonstration that BinJ/ZIKA-prME does not cause ADE of DENV infection and disease in the IFNAR^−/−^ mouse model, is perhaps encouraging rather than definitive, given the use of IFNAR^−/−^ mice and weight loss as a surrogate for severe disease. ADE may be associated with reduced type I IFN responses [52,53,54,55], a manifestation potentially difficult to discern in IFNAR^−/−^ mice; nevertheless, severe DENV pathology and ADE has been independently demonstrated in type I IFN response-deficient mice [56,57]. Severe DENV disease is also usually associated with vascular leakage and/or hemorrhage [58], whereas only weight loss was demonstrated in our model [19]. When clinical correlates of DENV ADE by anti-ZIKV antibodies are better characterized [24], it may become easier to design appropriate preclinical systems that can provide more definitive assurances.

This study supports the potential of BinJ/ZIKA-prME as a ZIKV vaccine to prevent CZS. The vaccine (at least in mice) does not appear to require adjuvant and is unable to replicate in the vaccine recipients [10], two attractive features for a vaccine where the intended target population includes pregnant women and women of child bearing age. 

## 5. Conclusions

Herein we provide preclinical data supporting the development of BinJ/ZIKA-prME as a ZIKV vaccine to prevent CZS. BinJ/ZIKA-prME is unable to produce progeny in the vaccine recipient, and BinJ/ZIKA-prME vaccination without the use of adjuvant was able to protect IFNAR^−/−^ mice against fetal brain infection. A multiplication-defective vaccine that does not require adjuvant represent attractive characteristics for a vaccine intended for use in pregnant women or women of child bearding age. A potential safety concern for ZIKV vaccines is ADE of DENV infection, which we show herein was not evident after BinJ/ZIKA-prME vaccination of IFNAR^−/−^ mice.

## Figures and Tables

**Figure 1 vaccines-08-00496-f001:**
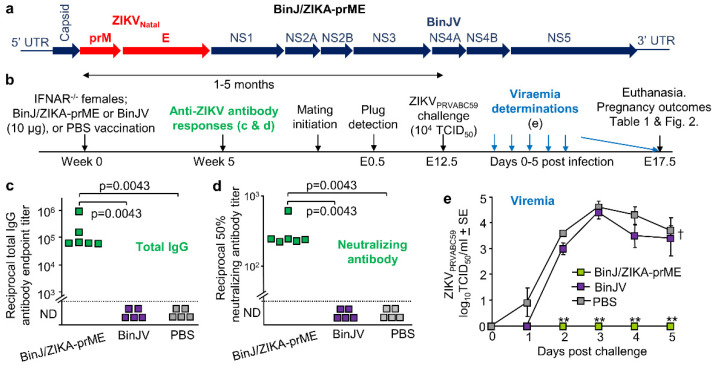
Vaccination, antibody assays, and ZIKV challenge. (**a**) Schematic illustration of gene arrangements for the chimeric BinJ/ZIKA-prME vaccine. (**b**) Timeline of study. The time between vaccination and challenge for the BinJ/ZIKA-prME vaccinated mice was 2, 5, 3, 2, 2, and 4 months (mean 3.2 ± SD 1.4 months) and for the BinJV vaccinated mice was 2, 5, 4, 2, and 1 months (mean 2.7 ± SD 1.7 months). (**c**) Endpoint anti-ZIKV IgG ELISA titers for mice vaccinated with BinJ/ZIKA-prME (*n* = 6), BinJV (*n* = 5) or PBS (*n* = 5). Limit of detection was 1 in 100. Statistics by Kolmogorov-Smirnov tests. (**d**) Anti-ZIKV 50% neutralizing antibody titers for serum samples described in c. Limit of detection was 1 in 30. Statistics as in c. (**e**) Mean viremia after s.c. challenge with ZIKV_PRVABC59_ (mouse numbers as in c). The limit of detection was 2 log_10_CCID_50_/mL per mouse. Statistics by Kolmogorov-Smirnov tests (** *p* = 0.0022).

**Figure 2 vaccines-08-00496-f002:**
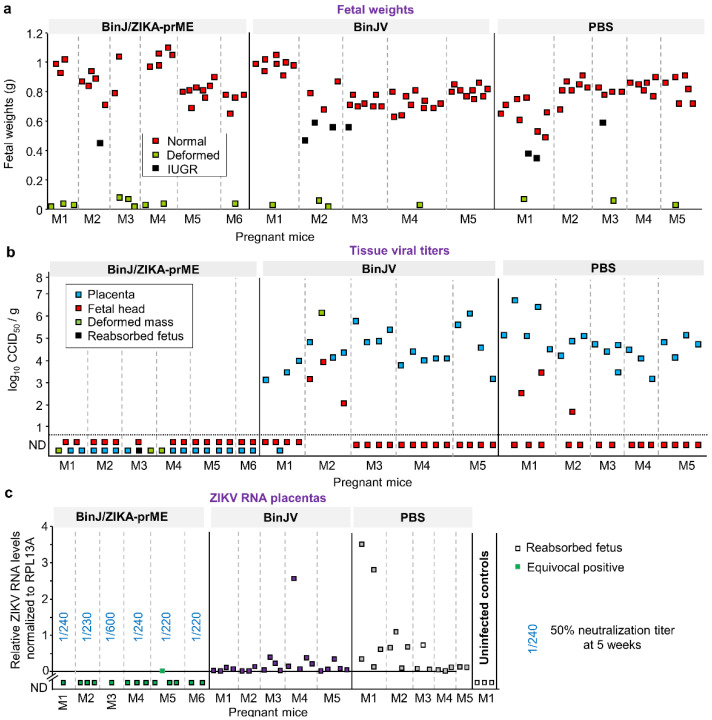
Fetal weights and viral loads in placental and fetal tissues. Mice were vaccinated with BinJ/ZIKA-prME, BinJV or PBS and at E12.5 dams were challenged with ZIKV_PRVABC59_ and tissues harvested at E17.5. (**a**) Fetal weights. Individual dams (pregnant mice) are indicated (for each vaccine group) on the x axis; each square representing one fetus. Vertical dashed grey lines separate litters from each dam. (**b**) Tissue ZIKV titers of indicated placental and fetal tissues. Limit of detection was 1.69 log_10_CCID_50_/g. (**c**) ZIKV RNA levels determined by qRT PCR in placenta. Three placenta from an unvaccinated and uninfected control dam were included as negative controls. The reabsorbed fetus indicates a placenta with no discernable fetus. The equivocal positive was tested in 11 repeat qPCR runs and had a Cq value <40 and a ZIKV-specific melt-curve for 5/11 repeat runs, and no Cq value and/or a non-specific melt curve for 6/11 runs. RNA levels in the BinJ/ZIKA-prME group were significantly lower than the two other groups *p* < 0.001 (Kolmogorov-Smirnov tests). Blue fractions for the BinJ/ZIKA-prME group indicate the serum dilution providing 50% ZIKV neutralization; serum was obtained at 5 weeks post vaccination (see Figure 1b). (**d**) ZIKV RNA levels in fetal heads. Statistics as for c, *p* < 0.001. (**e**) ZIKV RNA levels in deformed fetal/placental masses where fetus and placenta could not be distinguished.

**Figure 3 vaccines-08-00496-f003:**
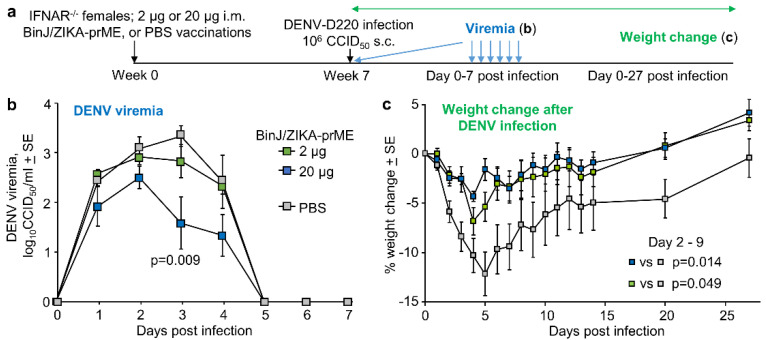
BinJ/ZIKA-prME vaccination and DENV ADE. (**a**) Time line of experiment (*n* = 6 mice per group). (**b**) Mean DENV-D220 viremias. Limit of detection 2 log_10_CCID_50_/mL per mouse. Statistics by Kruskal-Wallis test. (**c**) Mean percent weight change relative to day 0 after infection with DENV-D220. Statistics by repeat measure ANOVA for days 2 to 9 days post infection. Legend as in b.

**Table 1 vaccines-08-00496-t001:** Pregnancy outcomes at E17.5 after ZIKV challenge. Dam age was calculated from date of birth to E17.5. IUGR—intrauterine growth restriction.

Group	No. of Dams	Mean Number of Indicated Fetus Types + SE per Litter					Mean Dam Age
		Total	Normal	Reabsorbed	Deformed	IUGR	
BinJ/ZIKA-prME	6	6.3 ± 0.42	4.5 ± 0.85	0.2 ± 0.17	1.5 ± 0.56	0.2 ± 0.17	6.7 ± 0.56
BinJV	5	9 ± 0.55	7.4 ± 1.21	0	0.8 ± 0.37	0.8 ± 0.58	6.4 ± 0.93
PBS	5	7.6 ± 0.87	5.8 ± 0.49	0.2 ± 0.20	0.6 ± 0.24	1 ± 0.77	5 ± 0.84
Unvaccinated & uninfected	4	8.5 ± 0.50	7 ± 1.15	0	1.5 ± 0.96	0	3 ± 0.41

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
