# Peer review of "A Zika Vaccine Generated Using the Chimeric Insect-Specific Binjari Virus Platform Protects against Fetal Brain Infection in Pregnant Mice"

_vaccines, 2020, doi:10.3390/vaccines8030496_

Round 1

Reviewer 1 Report

Jessamine et al utilized an already published Binjari virus-based platform-chimeric flavivirus vaccine technology to produce Zika virus vaccine (BinJ/ZIKA-prME). To further validate its utility they utilized female mice where they vaccinated the mice challenged it with the ZIKVPRVABC59. They claimed no infection with Zika virus in maternal blood and placenta. Importantly, they utilized the model of DENV ADE and show limited cross-reactivity with the DENV virus which is important considering multiple abnormalities that arise due to autoimmune disorders.

The manuscript is well written, and authors have utilized sound technology. I have only a few minor comments

  1. Authors missed few important reports that need to be considered for discussion

A: https://www.nature.com/articles/s41467-018-02975-w

B: https://advances.sciencemag.org/content/6/32/eaba5068

2          In method sections, authors should mention the number of mice that are used

Overall, it is a good study supported by sound scientific data. The limitation is the absence of data in other animals such as Chimpanzee

Author Response

Point 1: Authors missed few important reports that need to be considered for discussion

A: https://www.nature.com/articles/s41467-018-02975-w

B: https://advances.sciencemag.org/content/6/32/eaba5068

Response 1: Both of the above references have now been included in this paper.

Reference A has now been incorporated into lines 45-50: “Live-attenuated chimeric flavivirus vaccines are well established and generally use an attenuated flavivirus vaccine as a backbone, with the latter’s pre-membrane and envelope (prME) replaced with the prME gene of the target flavivirus.  For instance, ZIKV prM with a licensed Japanese encephalitis live-attenuated vaccine as the backbone [8], or prME from dengue viruses with the licensed Yellow fever 17D live-attenuated vaccine as the backbone [9].  We recently applied this concept to the generation of chimeric flavivirus vaccines using Binjari virus (BinJV) [10].” This reference also appears as number 8 in the reference list.

Reference B has now been incorporated into line 42: “…or modified RNA vaccines…” This reference also appears as number 7 in the reference list.

Point 2:  In method sections, authors should mention the number of mice that are used

Response 2: Have altered the methods section under 2.4 ZIKV challenge of pregnant IFNAR-/- mice to include mouse numbers for the pregnancy model (lines 111 and 112). It now reads, “ZIKVPRVABC59 challenge of pregnant dams (n = 20) was undertaken as described [14] with some modifications.  Briefly, vaccinated (n = 6) and control (n = 4/5 per group) female IFNAR-/- mice…”

As to the number of mice for the ADE study, this is stated in line 137 “n=6 per group with a similar age distribution in each group)…”

Reviewer 2 Report

The manuscript "A Zika vaccine generated using the chimeric insect-specific Binjari virus platform protects against fetal brain infection in pregnant mice" describes additional assays to study the immunogenicity and protection efficacy provided by a recombinant Zika virus vaccine in IFNAR mice; the candidate vaccine has been obtained previously by replacing prM and E genes from Zika virus in the insect-specific Binjari flavivirus. The authors demonstrate convincingly that the candidate vaccine provides virological protection in dams and their fetuses, with no viral load evidenced by virus isolation nor by rtRT-PCR in blood, placenta and fetal tissues 1-5 days post-challenge. No fetal abnormalities were induced after ZIKV challenge in the control and immunized groups. Furthermore, anti-ZIKV immunity induced by the vaccination protocol afforded partial protection against DENV-2 challenge in IFNAR mice, with no indication for ADE during this ZIKV vaccination trial. 

Minor modifications are proposed

-Figure 1b, replace "ani-ZIK" by "anti-ZIKV antibody"

-As regards the methodology pertaining to Figure 1, the number of animals per group should be clearly indicated and the time elapsed between vaccination and challenge (for individuals and mean data) should be given. 

-Table 1, mean numbers of fetuses (total, normal, reabsorbed,...) should be presented per group, for a better comparison between the groups. 

-References 7 and 37 are identical

-LIne 222, (figure 2C)
